# Building Efficient ConvNets using Redundant Feature Pruning

**Babajide O. Ayinde & Jacek M. Zurada**
Department of Electrical and Computer Engineering
University of Louisville
Louisville, KY 40229, USA
{babajide.ayinde,jacek.zurada}@louisville.edu

## Abstract

This paper presents an efficient technique to prune deep and/or wide convolutional neural network models by eliminating redundant features (or filters). Previous studies have shown that over-sized deep neural network models tend to produce a lot of redundant features that are either shifted version of one another or are very similar and show little or no variations; thus resulting in filtering redundancy. We propose to prune these redundant features along with their connecting feature maps according to their differentiation and based on their relative cosine distances in the feature space, thus yielding smaller network size with reduced inference costs and competitive performance. We empirically show on select models and CIFAR-10 dataset that inference costs can be reduced by 40% for VGG-16, 27% for ResNet-56, and 39% for ResNet-110.

## 1 Introduction

Recent studies indicate that over-sized deep learning models typically result in largely over-determined (or over-complete) systems (Denil et al., 2013; Rodríguez et al., 2016; Bengio & Bergstra, 2009; Changpinyo et al., 2017; Ayinde & Zurada, 2017; Han et al., 2016; 2017). The resulting architectures may therefore be less computationally efficient due to their size, over-parameterization, and largely due to their high inference cost. To account for the scale, diversity and the difficulty of data these models learn from, the architectural complexity and the excessive number of weights and units are often deliberately built in into the deep neural network models by design (Bengio et al., 2007; Changpinyo et al., 2017). These over-sized models have expensive inference costs especially for applications with constrained computational and power resources such as web services, mobile and embedded devices. In addition to good accuracy, such resource-limited applications benefit greatly from lower inference cost (Li et al., 2017; Szegedy et al., 2016).

In this paper, we focus on controlled network size reduction of well-trained deep learning models based on feature agglomeration followed by *one-shot* elimination of redundant features and retraining heuristic. By leveraging on the observations that large capacity CNNs usually have significant redundancy among different features, we propose a simple, intuitive, and efficient way to remove such redundancy without undermining the efficiency or introducing sparsity that would require specialized library and/or hardware. We find that our pruning technique improves inference cost over a recently proposed technique (Li et al., 2017) across benchmark models and dataset considered without modifying existing hyperparameters.

## 2 Related Work

Storage and computational cost reduction via model network pruning techniques has a long history (LeCun et al., 1990; Hassibi & Stork, 1993; Mariet & Sra, 2016; Ioannou et al., 2016; Polyak & Wolf, 2015; Molchanov et al., 2017). For instance, Optimal Brain Damage (LeCun et al., 1990) and Optimal Brain Surgeon (Hassibi & Stork, 1993) use second-order derivative information of the loss function to prune redundant network parameters. Other related work include but is not limited to

Anwar et al. (2017) which prunes based on particle filtering, Mathieu et al. (2013) uses FFT to avoid overhead due to convolution operation, and Howard et al. (2017) uses depth multiplier method to scale down the number of filters in each convolutional layer. Closely related to our work, Li et al. (2017) sorts and prunes filters based on the sum of their absolute weights and Han et al. (2015) prunes weights with magnitude below a set threshold.

## 3 FEATURE CLUSTERING AND PRUNING

The objective here is to discover $n_f$ clusters in the set of $n'$ original filters that are identical or very similar in weight space according to a well-defined similarity measure, where $n_f \leq n'$. Achieving

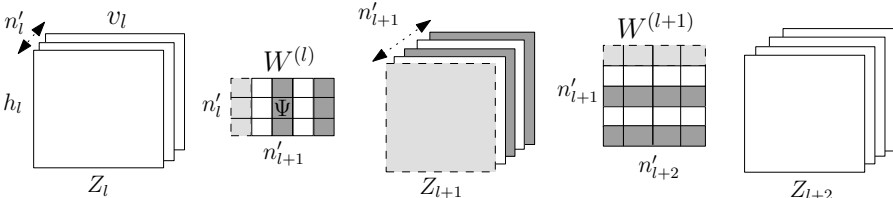

Figure 1: Pruning schema: Assume filter $\phi_1$ is the cluster representative, filters $\phi_3$ and $\phi_5$ and their corresponding feature maps in $Z_{1+1}$ and related weights in the next layer (third and fifth rows of $\mathbf{W}^{(l+1)}$) are pruned due to high similarity among filters $\phi_1$, $\phi_3$ and $\phi_5$. Filters $\phi_1$, $\phi_3$, and $\phi_5$ correspond to first, third, and fifth columns of $\mathbf{W}^{(l)}$, respectively

this involves choosing suitable similarity measures to express the inter-feature distances between features $\phi_i$ that connect the feature map $Z_{l-1}$ of layer $l-1$ to neurons of layer $l$. In other words, $\phi_i^{(l)}$, i=1,...$n_l'$, are feature vectors in layer $l$, each $\phi_i^{(l)} \in \mathbb{R}^p$ corresponds to the $i$-th column of the kernel matrix $\mathbf{W}^{(l)} = [\phi_1^{(l)}, \quad ...\phi_{n_l'}^{(l)}] \in \mathbb{R}^{p \times n_l'}$ where $p = k^2 n_{l-1}'$ and $k$ is the size of square 2D kernel $\Psi \in \mathbb{R}^{k \times k}$. A number of suitable agglomerative similarity testing/clustering algorithms can be applied for localizing redundant features. Based on a comparative review, a clustering approach from Walter et al. (2008); Ding & He (2002) has been adapted and reformulated for this purpose. By starting with each weight vector $\phi_i$ as a potential cluster, agglomerative clustering is performed by merging the two most similar clusters $C_a$ and $C_b$ as long as the average similarity between their constituent feature vectors is above a chosen cluster similarity threshold denoted as $\tau$ (Leibe et al., 2004; Manickam et al., 2000). The pair of clusters $C_a$ and $C_b$ exhibits average mutual similarities as follows:

$$\overline{SIM_C}(C_a, C_b) = \frac{\sum_{\phi_i \in C_a, \phi_j \in C_b} SIM_C(\phi_i, \phi_j)}{|C_a| \times |C_b|} > \tau$$
$$a, b = 1, ...n_l'; \ a \neq b; \ i = 1, ...|C_a|;$$
$$j = 1, ...|C_b|; \quad and \quad i \neq j$$

(1)

where $SIM_C(\phi_1, \phi_2) = \frac{<\phi_1, \phi_2>}{\|\phi_1\| \|\phi_2\|}$ is the cosine similarity between two features and $< \phi_1, \phi_2 >$ is the inner product of arbitrary feature vectors $\phi_1$ and $\phi_2$, and $\tau$ is a set threshold.

The redundant-feature-based pruning procedure for $l^{th}$ convolutional layer is summarized as follows:

1. Group all the filters $\phi_i$ (columns of the kernel matrix) into $n_f$ clusters whose average similarities are above a set threshold $\tau$.

2. Two heuristics are considered: (A) Randomly sample one representative filter from each of the $n_f$ clusters and prune the remaining filters and their corresponding feature maps. (B) Randomly prune $n' - n_f$ filters and their corresponding feature maps. The weights of the pruned feature maps in $l^{th}$ layer are also removed in layer $(l+1)^{th}$ as shown in Figure 1.

3. A new kernel matrix is defined for both $l^{th}$ and $(l+1)^{th}$ layer of a new smaller model.

## 4 EXPERIMENTS

The network pruning was implemented in Pytorch deep learning library (Paszke et al., 2017). We evaluated the proposed redundant-feature-based pruning on three deep networks, namely: VGG-16 (Simonyan & Zisserman, 2015) and two residual networks (ResNet-56 and 110) (He et al., 2016) trained on CIFAR-10. The baseline model and accuracy for residual networks were obtained by training the model following the procedures highlighted in He et al. (2016). See Appendix for implementation details and supplemental results for both VGG-16 and residual networks.

| Model | Error % | FLOP | Pruned % | # Parameters | Pruned % |
|---|---|---|---|---|---|
| VGG-16 | 6.20 | $3.13 \times 10^8$ | | $1.47 \times 10^7$ | |
| VGG-16-pruned (Li et al., 2017) | 6.60 | $2.06 \times 10^8$ | 34.2% | $5.4 \times 10^6$ | 64.0% |
| **VGG-16-pruned-A (this work)** | **6.33** | $\mathbf{1.86 \times 10^8}$ | **40.5%** | $\mathbf{3.23 \times 10^6}$ | **78.1%** |
| VGG-16-pruned-B (this work) | 6.70 | $1.86 \times 10^8$ | 40.5% | $3.23 \times 10^6$ | 78.1% |
| ResNet-56 | 6.61 | $1.25 \times 10^8$ | | $8.5 \times 10^5$ | |
| ResNet-56 pruned (Li et al., 2017) | 6.94 | $9.09 \times 10^7$ | 27.6% | $7.3 \times 10^5$ | 13.7% |
| **ResNet-56 pruned-A (this work)** | **6.88** | $\mathbf{9.07 \times 10^7}$ | **27.9%** | $\mathbf{6.5 \times 10^5}$ | **23.7%** |
| ResNet-56 pruned-B (this work) | 6.94 | $9.07 \times 10^7$ | 27.9 % | $6.5 \times 10^5$ | 23.7 % |
| ResNet-110 | 6.35 | $2.53 \times 10^8$ | | $1.72 \times 10^6$ | |
| ResNet-110 pruned (Li et al., 2017) | **6.70** | $1.55 \times 10^8$ | 38.6% | $1.16 \times 10^6$ | 32.4% |
| **ResNet-110 pruned-A (this work)** | 6.73 | $\mathbf{1.54 \times 10^8}$ | **39.1%** | $\mathbf{1.13 \times 10^6}$ | **34.2%** |
| ResNet-110 pruned-B (this work) | 7.41 | $1.54 \times 10^8$ | 39.1% | $1.13 \times 10^5$ | 34.2% |

Table 1: Performance evaluation for three pruning techniques on CIFAR-10 dataset. Performance with the lowest test error is reported.

### 4.1 VGG-16 ON CIFAR-10

As seen in Table 1, for $\tau = 0.54$ our approaches (both A and B) outperform that in Li et al. (2017) and are able to prune more than $78\%$ of the parameters resulting in $40\%$ FLOP reduction and a competitive classification accuracy. We suspect that our pruning approach outshines that of Li et al. (2017), which ranks importance of filters based on the sum of absolute value of their weights, because it localizes and prunes similar or shifted versions of filters that do not add extra information to the feature hierarchy. This notion is reinforced from information theory standpoint that the activation of one unit should not be predictable based on the activations of other units of the same layer (Rodríguez et al., 2017). Another crucial observation is that heuristic A achieves a better accuracy than heuristic B because random pruning might prune filters that are dissimilar.

### 4.2 RESNET-56/110 ON CIFAR-10

For RestNet-56/110, we only prune the first layer of the residual block to avoid dimensions mismatch due to unavailability of projection mapping for selecting the identity mapping (see He et al. (2016) for details). We found that redundant-feature-based pruning are competitive to that in Li et al. (2017) in terms of $\%$ FLOP reduction. However, for RestNet-56, our approach prunes $10\%$ more parameters than Li et al. (2017) and upon retraining, it achieves better classification accuracy.

## 5 CONCLUSION

Motivated by the observations of recent studies that modern CNNs often have large number of overlapping filters amounting to unnecessary filtering redundancy and large inference cost. By using hierarchical agglomerative clustering to group all filters at each layer in the weight space according to a predefined measure, redundant filters are pruned and inference cost (FLOPS) reduced by $40\%$ for VGG-16, $28\%/39\%$ for ResNet-56/110 trained on CIFAR-10. To recover the accuracy after pruning, we retrained the model for a few iterations without the need to modify hyper-parameters.

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

ACKNOWLEDGMENTS

This work was supported by the NSF under grant 1641042.

APPENDIX

## 6 IMPLEMENTATION DETAILS AND RESULTS

All experiments were performed on Intel(r) Core(TM) i7-6700 CPU @ 3.40Ghz and a 64GB of RAM running a 64-bit Ubuntu 14.04 edition. The software implementation has been in Pytorch library [1] on two Titan X 12GB GPUs and the filter clustering was implemented in SciPy ecosystem Jones et al.. The agglomeration of filters using hierarchical clustering is practical for very wide and deep networks even though its complexity is $O((n_l')^2\log(n_l'))$. For instance, clustering VGG-16 feature vectors empirically takes on the average on our machine 14.1 milliseconds and this is executed only once during training. This amounts to a negligible computational overhead for most deep architectures.

The implementation of our filter pruning strategy is similar to that in Li et al. (2017) in the sense that when a particular filter of a convolutional layer is pruned, its corresponding feature map is also pruned and the weights of the pruned feature map in the filter of the next convolutional layer are equally pruned. It must be emphasized that after pruning the feature maps of last convolutional layer, the input to the linear layer has changed and its weight matrix has to be pruned accordingly.

---

[1]`http://pytorch.org/`

CIFAR-10 dataset was used in all the experiments to train and validate the models. The dataset contains a labeled set of 60,000 32x32 color images belonging to 10 classes: airplanes, automobiles, birds, cats, deer, dogs, frogs, horses, ships, and trucks. The dataset is split into 50000 and 10000 training and testing sets, respectively. We used FLOP to compare the computational efficiency of the models considered because its evaluation is independent of the underlying software and hardware. In order to fairly compare our method with that in Li et al. (2017), we also calculated the FLOP only for the convolution and fully connected layers.

## 6.1 VGG-16 ON CIFAR-10

For the first set of experiments, we used a modified version of the popular convolutional neural network known as the VGG-16 (Simonyan & Zisserman (2015)), which has 13 convolutional layers and 2 fully connected layer. In the modified version of VGG-16, each layer of convolution is followed by a Batch Normalization layer (Ioffe & Szegedy, 2015). Our base model was trained for 350 epochs, with a batch-size of 128 and a learning rate 0.1 as highlighted in the repository[2]. The learning rate was reduced by a factor of 10 at 150 and 250 epochs. We have shared our pruning implementation and trained model for reproducibility of results [3]. After pruning we retrain the network with learning rate of 0.001 for 80 epochs to fine tune the weights of the remaining connections to regain the accuracy.

Figure 2 shows the number of nonredundant filters per layer for different $\tau$ values. As can be seen

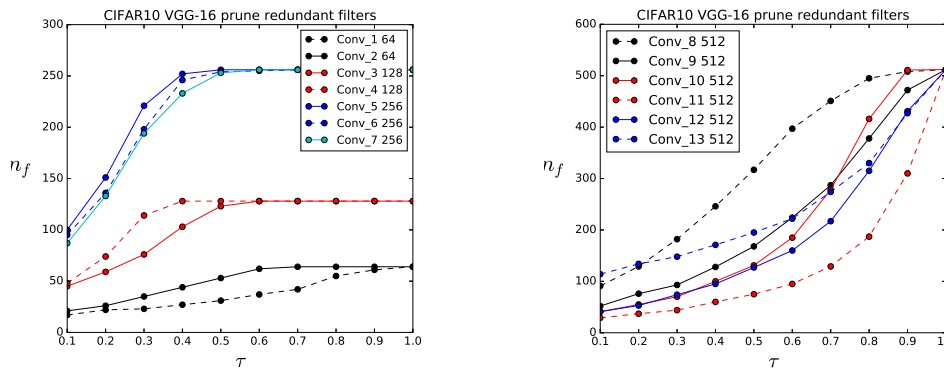

Figure 2: Number of nonredundant filters ($n_f$) vs. cluster similarity threshold ($\tau$) for VGG-16 trained on the CIFAR-10 dataset. Initial number of filters for each layer is shown in the legend.

that some convolutional layers in VGG are prone to extracting features with very high correlation such as Conv layer 1, 11, 12, and 13. Another very important observation is that later layers of VGG are more susceptible to extracting redundant filters than earlier layers and can be pruned heavily. Figure 3(a) shows the sensitivity of the convolutional layer of VGG-16 to pruning and it can be observed that layers such as Conv 1, 3, 4, 9, 11, and 12 are very sensitive. However, as can be observed in Figure 3(c), accuracy can be restored after pruning filters in later layers (Conv 9, 11, and 12) compared to early ones (Conv 1, 3, and 4). For our final test score, we fine tuned on the entire training set. For pruning, we performed a grid search over $\tau$ values within 0.1 and 1.0, and found 0.54 gave the least test error. Table 2 reports the pruning performance for $\tau = 0.54$ and it can be easily seen that more than 90% of most of the latter layers have been pruned and most of the sensitive earlier layers are minimally pruned. Figure 3(b) depicts the sensitivity of trained VGG-16 model to pruning using heuristic B that calculates the number of redundant filters ($n' - n_f$) and randomly prunes them.

---

[2]Implementation of modified version of VGG-16 can be found in `https://github.com/kuangliu/pytorch-cifar`

[3]`https://github.com/babajide07/Redundant-Feature-Pruning-Pytorch-Implementation`

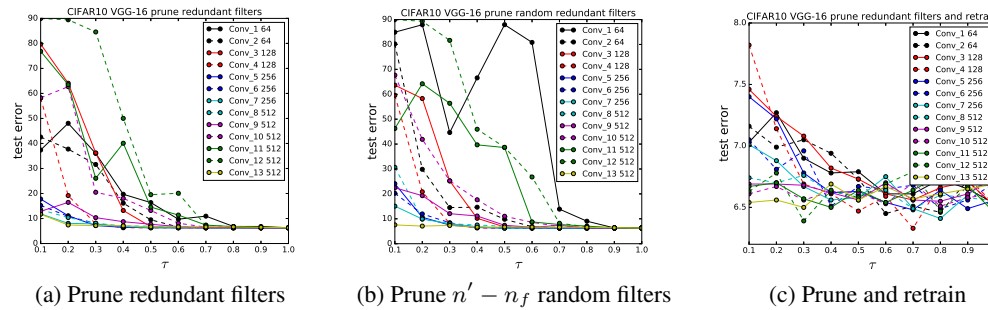

| (a) Prune redundant filters | (b) Prune $n' - n_f$ random filters | (c) Prune and retrain |

Figure 3: Sensitivity to pruning (a) redundant filters (b) random $n' - n_f$ filters, and (c) redundant filters and retraining for 30 epochs for VGG-16.

| layer | $v_l \times h_l$ | #Maps | FLOP | #Params | #Maps | FLOP% |
|---|---|---|---|---|---|---|
| Conv_1 | $32 \times 32$ | 64 | 1.8E+06 | 1.7E+03 | 32 | 50.0% |
| Conv_2 | $32 \times 32$ | 64 | 3.8E+07 | 3.7E+04 | 58 | 54.7% |
| Conv_3 | $16 \times 16$ | 128 | 1.9E+07 | 7.4E+04 | 125 | 11.5% |
| Conv_4 | $16 \times 16$ | 128 | 3.8E+07 | 1.5E+05 | 128 | 2.3% |
| Conv_5 | $8 \times 8$ | 256 | 1.9E+07 | 2.9E+05 | 256 | 0% |
| Conv_6 | $8 \times 8$ | 256 | 3.8E+07 | 5.9E+05 | 254 | 0.8% |
| Conv_7 | $8 \times 8$ | 256 | 3.8E+07 | 5.9E+05 | 252 | 2.3% |
| Conv_8 | $4 \times 4$ | 512 | 1.9E+07 | 1.2E+06 | 299 | 42.5% |
| Conv_9 | $4 \times 4$ | 512 | 3.8E+07 | 2.4E+06 | 164 | 81.3% |
| Conv_10 | $4 \times 4$ | 512 | 3.8E+07 | 2.4E+06 | 121 | 92.4% |
| Conv_11 | $2 \times 2$ | 512 | 9.4E+06 | 2.4E+06 | 59 | 97.3% |
| Conv_12 | $2 \times 2$ | 512 | 9.4E+06 | 2.4E+06 | 104 | 97.7% |
| Conv_13 | $2 \times 2$ | 512 | 9.4E+06 | 2.4E+06 | 129 | 94.9 % |

Table 2: Pruning performance on CIFAR dataset using VGG-16 model at $\tau = 0.54$

## 6.2 RESNET-56/110 ON CIFAR-10

The architecture of residual networks is more complex than VGG and also the number of parameters in the fully connected layer is relatively smaller and this makes it a bit challenging to prune a large proportion of the parameters. Both ResNet-56 and ResNet-110 have three stages of residual blocks for feature maps with of differing sizes. The sizes ($v_l \times h_l$) of feature maps in stages 1,2, and 3 are $32 \times 32$, $16 \times 16$, and $8 \times 8$, respectively. Each stage has 9 and 18 residual blocks for ResNet-56 and ResNet-110, respectively. A residual block consists of two convolutional layer each followed by a Batch Normalization layer. Preceding the first stage is a convolutional layer followed by a Batch Normalization layer[4]. Only the redundant filters in first convolution layer of each block are pruned due to unavailability of mapping for selecting the identity feature maps.

As can be observed in Figures 4 and 5 that convolutional layers in first stage are prone to extracting more redundant features than those of second stage, and the convolutional layers in the second stage are susceptible to extracting redundant filters than those of third block, which is contrary to the observations with VGG-16. In effect, more filters could be pruned from layers in first stage than the latter ones without losing much to accuracy. More specifically, many layers in the first stage of ResNet-56, such as Conv 2,8,10, and 26, have filters that are correlated more $80\%$ and could be heavily pruned. Similarly, convolutional layers in the first stage of ResNet-110 exhibit similar tendency to produce more filters that are redundant. As a result of these differing tendencies at each stage, $\tau$ for all the stages is set to different values. In pruning ResNet-56, we set $\tau$ to $0.253$, $0.223$, $0.20$ as thresholds for stages 1,2, and 3, respectively. Similarly for ResNet-110 we used $0.18$, $0.12$, and $0.17$.

---

[4]We used the Pytorch implementation of ResNet56/110 in `https://github.com/D-X-Y/ResNeXt-DenseNet` as baseline models

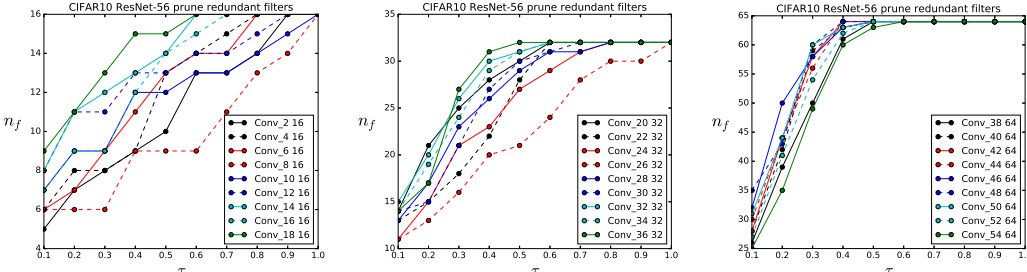

Figure 4: Number of nonredundant filters ($n_f$) vs. cluster similarity threshold ($\tau$) for ResNet-56 trained on the CIFAR-10 dataset. Initial number of filters for each layer is shown in the legend.

Figure 6 shows the sensitivity of the convolutional layer of ResNet-56 to pruning and it can

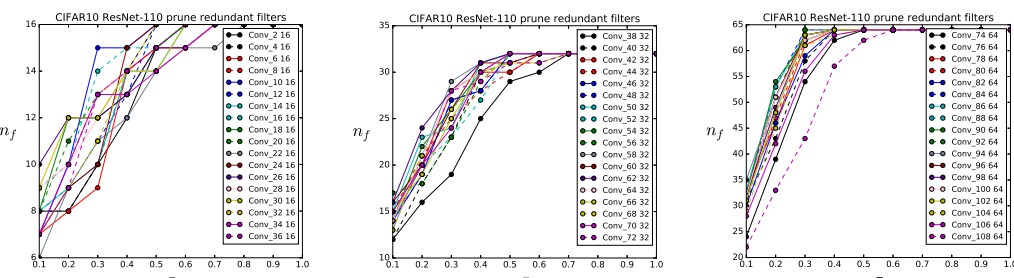

Figure 5: Number of nonredundant filters ($n_f$) vs. cluster similarity threshold ($\tau$) for ResNet-110 trained on the CIFAR-10 dataset. Initial number of filters for each layer is shown in the legend.

be observed that layers such as Conv 10, 14, 16, 18, 20, 34, 36, 38, 52 and 54 are more sensitive to filter pruning than other convolutional layers. Likewise for ResNet-110, the sensitivity of the convolutional layer to pruning is depicted in Figure 7 and it can be observed that Conv 1, 2, 38, 78, and 108 are sensitive to pruning. In order to regain the accuracy by retraining the pruned model, we skip these sensitive layers while pruning.

As seen on Table 1 for ResNet-56, redundant-feature-based pruning (both A and B) have

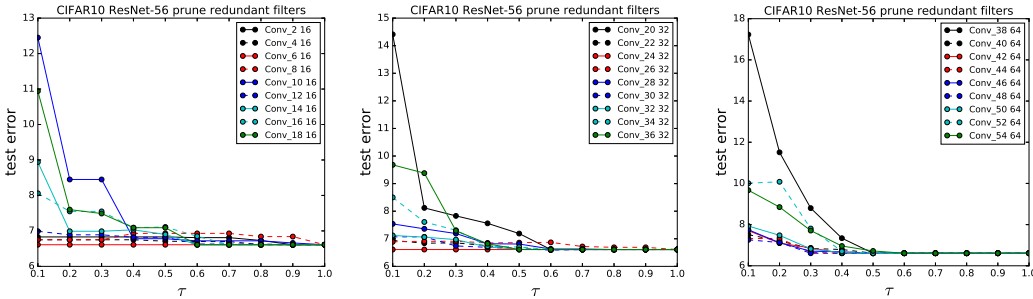

Figure 6: Sensitivity to pruning $n' - n_f$ redundant convolutional filters in ResNet-56

competitive performance in terms of FLOP reduction but outperform that in Li et al. (2017) in reducing the number of effective parameters by 10% with relatively better classification accuracy after retraining. However, we were able to marginally increase the effective number of parameters pruned in ResNet-110 from 38.6% to 39.1%, which gives rise to approximately 2% increase. Also, the accuracy after fine tuning the pruned ResNet-110 model is not as good as that in Li et al. (2017).

The inference time of both original and pruned models was recorded and reported in Ta-

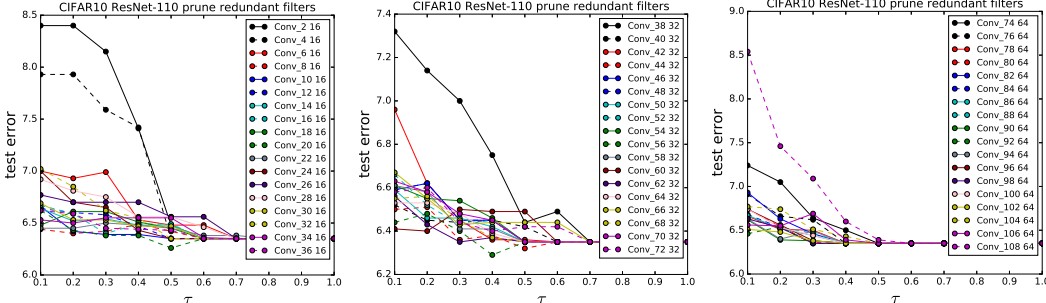

Figure 7: Sensitivity to pruning $n' - n_f$ redundant convolutional filters in ResNet-110

| Model | FLOP | Pruned % | Time(s) | Saved % |
|---|---|---|---|---|
| VGG-16 | $3.13 \times 10^8$ | | 1.47 | |
| VGG-16-pruned-A (this work) | $1.86 \times 10^8$ | 40.5% | 0.94 | 34.01% |
| ResNet-56 | $1.25 \times 10^8$ | | 1.16 | |
| ResNet-56-pruned-A (this work) | $9.07 \times 10^7$ | 27.9% | 0.96 | 17.2% |
| ResNet-110 | $2.53 \times 10^8$ | | 2.22 | |
| ResNet-110-pruned-A (this work) | $1.54 \times 10^8$ | 39.1% | 1.80 | 18.9% |

Table 3: FLOP and wall-clock time reduction for inference. Operations in convolutional and fully connected layer are considered for computing FLOP

ble 3. 10000 test images of CIFAR-10 dataset were used for the timing evaluation conducted in Pytorch version 0.2.0_3 with Titan X (Pascal) GPU and cuDNN v8.0.44, using a mini-batch of size 100. It can be observed that %FLOP reduction also translates almost directly into inference clock time savings.

## 6.3 PRUNE AND TRAIN FROM SCRATCH

In order to see the effect of copying weights from the original (larger) model to a pruned (smaller) model, we pruned two models (VGG-16 and ResNet-56) as described above and re-initialized their weights and trained them from scratch. As shown in Table 4 that fine tuning a pruned model is almost always better than re-initializing and training a pruned model from scratch. We believe that already-trained filters may serve as good initialization for a smaller network which might on its own be difficult to train. Other observation from Table 4 is that redundant-feature-based pruning results in an architecture that when trained attains a better performance than its counterpart in Li et al. (2017). This may indicate that redundant-feature-based pruning might be a potential approach to determining the architectural width of modern deep neural network models.

| Model | Error % |
|---|---|
| VGG-16-pruned (Li et al., 2017) | 6.60 |
| VGG-16-pruned-A (this work) | 6.33 |
| VGG-16-pruned-scratch-train (Li et al., 2017) | 6.88 |
| VGG-16-pruned-A-scratch-train (this work) | 6.79 |
| ResNet-56-pruned (Li et al., 2017) | 6.94 |
| ResNet-56-pruned (this work) | 6.88 |
| ResNet-56-pruned-scratch-train (Li et al., 2017) | 8.69 |
| ResNet-56-pruned-A-scratch-train (this work) | 7.66 |

Table 4: Performance on CIFAR dataset

