# OpenReview forum: "Building Efficient ConvNets using Redundant Feature Pruning"
_ICLR.cc/2018/Workshop — Reject_

### Official Review · AnonReviewer2 · 2018-03-10
**Agglomerative clustering to prune cnn filters for efficient evaluation. Preliminary work well behind the state-of-the-art. Reject.**

**Rating:** 4
**Confidence:** 4

**Review:**

The authors propose the use of agglomerative clustering to group and then subsequently sample cnn filters to remove redundancy and speed up the test-time execution of convolutional networks. They show that their approach can be used to speed up VGG and ResNets trained on on CiFar10 by ~25-40% with small degradation (about 5% relative), outperforming a technique recently proposed by Li et al., which prunes cnn filters based on the sum of their absolute weights.

Their technique, while intuitely appealing and realitively simple to implement, is not established as state-of-the-art. Several important, highly related, and more sophisticated recent papers on efficient evalution of cnns using filter clustering & subspace analysis are not cited or compared to qualitatively or quantitatively (see below). These existing techniques can speed up evalutions by 200% or more with negligible degradation (e.g. [4r] below).

Overall, the paper does not contain any new, ground-breaking ideas or results, and is in a preliminary state. As such, the paper falls well below the acceptance threshold for ICLR. Reject.

Some important & highly related recent papers on the efficient evaluation of cnns:

[1r] Denton, Emily L., Wojciech Zaremba, Joan Bruna, Yann LeCun, and Rob Fergus. "Exploiting linear structure within convolutional networks for efficient evaluation." In Advances in neural information processing systems, pp. 1269-1277. 2014.
-not cited

[2r] Han, Song, Huizi Mao, and William J. Dally. "Deep compression: Compressing deep neural networks with pruning, trained quantization and huffman coding." arXiv preprint arXiv:1510.00149 (2015). (ICLR 2016)
-in bib but not cited

[3r] Han, Song, Jeff Pool, John Tran, and William Dally. "Learning both weights and connections for efficient neural network." In Advances in neural information processing systems, pp. 1135-1143. 2015.
-in bib but not cited

[4r] Jaderberg, Max, Andrea Vedaldi, and Andrew Zisserman. "Speeding up convolutional neural networks with low rank expansions." arXiv preprint arXiv:1405.3866 (2014).
https://arxiv.org/pdf/1405.3866.pdf

---

### Official Review · AnonReviewer3 · 2018-03-10
**Pruning redundant filters via clustering is not a new idea**

**Rating:** 6
**Confidence:** 3

**Review:**

Pruning CNN models has been heavily studied in the recent years.   This work presents a new method for pruning based redundancy/similarity. The results show that it compares favorably to Li et al's structural pruning method based on L1 norm, which is encouraging.
However, it should be noted that the idea of using filter similarity for pruning has been previously explored by RoyChowdhury et al. in their paper "Reducing Duplicate Filters in Deep Neural Networks". The only difference is that here hierarchical clustering is used to decide redundancy whereas RoyChowdhury et al.'s work uses a simple threshold of similarity to decide what are considered to be "duplicate".  This particular work was missing from the paper's discussion and comparison.

---

### Official Review · AnonReviewer1 · 2018-03-11
**clustering and pruning redundant filters in neural networks for faster inference**

**Rating:** 6
**Confidence:** 4

**Review:**

The paper proposes to cluster the filters in each layer of a neural network based on cosine similarities and eliminate redundant filters, i.e., filters that belong to the same cluster. They show their approach outperforms alternative filter pruning methods based on magnitude if weight kernels.

Q: Could the authors clarify what they mean by: "A new kernel matrix is defined for both lth and (l + 1)th layer of a new smaller model"
If we remove feature channels from a set of feature maps at layer l, since the kernels in the layer above expect all of them present, how do we handle that? Do you mean that we use the cluster representative everywhere instead? I believe I am missing something obvious, but please do clarify. Figure 1 was not very useful either for me to understand.

---

### Decision · Program_Chairs · 2018-03-20
**ICLR 2018 Workshop Acceptance Decision**

**Decision:**

Reject

**Comment:**

Based on the reviews, this paper has not been accepted for presentation at the ICLR workshop. However, the conversation and updates can continue to appear here on OpenReview.